# Low-Dose Naltrexone as an Adjuvant in Combined Anticancer Therapy

**DOI:** 10.3390/cancers16061240

**Published:** 2024-03-21

**Authors:** Marianna Ciwun, Anna Tankiewicz-Kwedlo, Dariusz Pawlak

**Affiliations:** Department of Pharmacodynamics, Medical University of Bialystok, Mickiewicza 2C, 15-222 Bialystok, Poland; anna.tankiewicz-kwedlo@umb.edu.pl (A.T.-K.); dariusz.pawlak@umb.edu.pl (D.P.)

**Keywords:** naltrexone, low-dose naltrexone, cancer, synergistic therapy, adjuvant

## Abstract

**Simple Summary:**

This review aims to present current evidence on the potential use of low-dose naltrexone (LDN) in cancer therapy. Low-dose naltrexone exhibits an inhibitory effect on cancer cell proliferation by blocking the opioid growth factor–receptor axis and enhancing the immune response against cancer cells. Data from existing studies indicate that low-dose naltrexone has a high anti-cancer potential, especially as an adjuvant in conventional chemotherapy and immunotherapy schemes.

**Abstract:**

Naltrexone (NTX) is a non-selective antagonist of opioid receptors, primarily used in the therapy of opioid and alcohol dependence. Low-dose naltrexone (LDN) exhibits antagonistic action against the opioid growth factor receptor (OGFr), whose signaling is associated with the survival, proliferation, and invasion of cancer cells. The mechanism of action of LDN depends on the dose and duration of the OGFr blockade, leading to a compensatory increase in the synthesis of the opioid growth factor (OGF), which has an inhibitory effect on carcinogenesis. Numerous studies on in vitro and in vivo models provide evidence of LDN’s positive impact on inhibiting the OGF–OGFr axis in cancers. LDN’s unique mechanism of action on cancer cells, lack of direct cytotoxic effect, and immunomodulating action form the basis for its use as an adjuvant in chemotherapy and immunotherapy of cancerous lesions.

## 1. Introduction

Cancer pain is a serious issue that constitutes a significant barrier to effective therapy and the daily functioning of oncological patients. According to data from 2016, 55% of patients undergoing cancer treatment and 66% of those with advanced cancer suffer from chronic pain, which translates to worse prognoses for the patients and a decrease in the effectiveness of the applied anti-cancer treatment [1]. There are schemes for effective pharmacotherapy of pain using both non-opioid and opioid medications. New literature data suggest that the use of opioids may be associated with a worse response to the applied pharmacotherapy, while simultaneously exacerbating cancer progression [2]. In recent years, it has been discovered that many cancers exhibit overexpression of opioid receptors, which are target points for classical opioid drugs, and stimulating their activity is linked to increased carcinogenesis [3]. Moreover, the discovery of the new ζ-opioid receptor and the OGF–OGFr axis has posed the question to the scientific community of whether the use of opioid receptor antagonists, such as naltrexone and methylnaltrexone, can impact the reversal of the negative effects of using classical opioids in cancer therapy [4].

In this literature review, we presented research on the use of naltrexone and methylnaltrexone (MNTX) in the context of enhancing anticancer effects. Both of these compounds belong to opioid receptor antagonists, but due to their chemical structure (quaternary amine), MNTX does not penetrate the blood barrier and therefore its action is limited to peripheral receptors.

The available literature data, supported by the results of in vitro and in vivo studies, indicate the potential use of LDN as an adjuvant in combined anticancer therapy. The mechanism of this beneficial effect is not clear. It is the result of the influence on the opioid growth factor receptor (OGFr) axis, which results in reduced cell replication and an increase in the cytolytic activity of NK cells, as well as stimulation of INF-γ and IL-2 production. The existence of other additional mechanisms of action cannot be ruled out; therefore, it is necessary to thoroughly understand the biological effects of naltrexone.

## 2. The µ Receptor and Cancer

Opioid medications commonly prescribed for cancer-related pain, such as morphine, fentanyl, and oxycodone, exert a strong analgesic effect through agonistic action on µ-opioid receptors [5]. µ receptors are primarily located in the neurons of the central nervous system, where their stimulation is mainly associated with analgesia. Meanwhile, µ receptors present in immune system cells and vascular endothelium are responsible for immunosuppressive effects and angiogenesis [6,7]. It has also been shown that cancer cells and structures forming the tumor microenvironment express an overexpression of µ receptors compared to normal cells, which is particularly alarming, as recent years have seen an increasing number of reports on poorer patient prognoses who receive opioid medication therapy for cancer pain [8]. Studies have confirmed that activation of this receptor can initiate and regulate numerous cellular responses, such as increased proliferation, survival, migration, and tumor invasion [9]. Morphine-induced activation of the µ-opioid receptor led to an increase in the expression of the epidermal growth factor receptor (EGFR), which caused an increase in proliferation in vitro of human non-small cell lung cancer (NSCLC) cell lines through phosphorylation of MAPK, ERK, and protein kinase B, which are responsible for triggering the synthesis of proteins involved in the proliferation, migration, and epithelial-mesenchymal transition of cancer cells [10]. Numerous literature data provide evidence of µ receptor overexpression in cancers, such as negative breast cancer [2], pancreatic cancer [11], colon cancer [12], esophageal squamous cell carcinoma [13,14] and laryngeal cancer [15], stomach cancer [16], hepatocellular carcinoma [17], prostate cancer, and lung cancer [18]. In addition to the increased expression of µ-opioid receptors in numerous types of cancers, an increase in the expression of another target point, namely the ζ-opioid receptor, has also been observed in these cells, whose signaling is closely related to the proliferation and survival of cells that have undergone carcinogenesis [19].

## 3. The OGF–OGFr Axis

The opioid growth factor (met-enkephalin; OGF) is a native pentapeptide belonging to the family of endogenous opioids. It is produced autocrinally and paracrinely by cells, and its target sites of action are both normal and dysfunctionally replicating tissues [20]. OGF plays a key role in a range of biological functions of the body as a neuroprotective agent, participates in tissue development and organ maturation, regulates DNA synthesis and angiogenesis, and contributes to tissue regeneration and wound healing processes [21]. OGF exhibits affinity to the opioid growth factor receptor (OGFr), described as the ζ-opioid receptor. The molecular and biochemical structure of OGFr significantly differs from the biological structure of the “classical” opioid receptors µ, κ, and γ [22]. The classical opioid receptors are 7-transmembrane, cytoplasmic receptors. The gene structures encoding them are homologous, and many ligands for µ, κ, and γ receptors bind across, interacting with more than one receptor. The sequencing of the genes encoding OGFr revealed that this receptor has negligible gene homology to the classical opioid receptors, yet it possesses stereospecificity towards ligands specific to other receptors [23,24,25]. Furthermore, unlike µ, γ, and κ receptors, OGFr is located on the outer membrane of the cell nucleus and inside the nucleus [26]. The OGF enters the cell via active transport, binding to the OGFr located on the outer nuclear membrane. It then moves inside the nucleus, where it stimulates the activity of cyclin-dependent kinase inhibitors p16/p21, resulting in the blocking of the G1/S phase of the cell cycle [27,28,29]. The OGF–OGFr axis has been identified in several types of cancer tissues, and the effects of its signaling are linked to the immunological modulation of cytokine-releasing cells, which can remodel the tumor microenvironment by enhancing anti-cancer activity and alleviating immunosuppressive action [30]. Moreover, the OGF inhibits the proliferation of cancer cells by arresting the cell cycle and/or inducing apoptosis [31]. The influence of the μ-opioid receptor and OGFr signaling in cancer cells is shown in Figure 1.

The presented data indicate that the impact of opioid receptor agonists on signaling and cell activity in the cancer niche cannot be ignored, and it is necessary to consider effective therapy that mitigates the negative impact of µ-opioid receptor stimulation and the OGF–OGFr axis.

## 4. Naltrexone

Naltrexone (NTX) is an opioid receptor antagonist with structural similarity to opioids. The mechanism of action of NTX is based on negating the effects of exogenous opioids through competitive binding with opioid receptors. It exhibits a strong affinity for µ-opioid receptors located in the central nervous system. Additionally, it has high affinity, but also partial agonistic activity towards κ receptors in the brain and spinal cord, and minor affinity for δ receptors present in the spinal cord and peripheral nervous system [32,33]. Studies using positron emission tomography have shown that NTX at a therapeutic dose of 50 mg saturates about 95% of µ-opioid receptors [34].

In 1984, NTX was approved by the U.S. Food and Drug Administration (FDA) for the treatment of alcohol dependence and opioid addiction [35]. Additionally, phase III clinical trials are ongoing for the use of NTX in an extended-release formulation combined with bupropion for obesity treatment [36].

### 4.1. Naltrexone Pharmacokinetics

Pharmacological data describing the safety profile of naltrexone reveal that its use at a dose of 300 mg daily may lead to liver cell damage [37]. However, naltrexone at a dose of 50–100 mg and lower is considered completely safe for humans. This is partly due to the poor bioavailability (5–40% due to the first-pass effect) of naltrexone after oral administration, which means that systemic side effects following this route of administration are minimal. In turn, parenteral administration of naltrexone may potentially lead to side effects [38]. Regarding LDN, data on the actual effects of the drug are still limited. Results from clinical trials indicate that all low-dose naltrexone, very low-dose naltrexone, and ultra-low-dose naltrexone are well tolerated, even with concurrent opioid therapy.

After parenteral administration, naltrexone is rapidly distributed in the body, easily crosses the placenta, and binds relatively poorly to albumin. Its metabolism takes place in the liver by dihydrodiol dehydrogenases into 6β-naltrexol (6β-hydroxynaltrexone) [39]. The biotransformation of naltrexone in the liver is individually variable in both children and adults, depending primarily on genetic variability, age, and sex [38]. There are indications that the AKR1C4 genotype has a large impact on the biotransformation of naltrexone. In men, due to the high concentration of testosterone and dihydrotestosterone, the formation of 6βN is inhibited. The literature data indicate that adults treated with oral naltrexone had greater than a 10-fold variability in systemic exposure (e.g., Cmax and area under the curve). According to some authors, the average half-lives of naltrexone and 6β-naltrexol were approximately 4 and 12 h, respectively [40]. According to others, the serum half-life of naltrexone in adults ranged from 30 to 81 min (mean 64 ± 12 min). In neonates, the mean plasma half-life was 3.1 ± 0.5 h. Naltrexone administered orally or intravenously is approximately 25 to 40% excreted renally as metabolites within 6 h, approximately 50% within 24 h, and 60 to 70% within 72 h [38].

The renal clearance of naltrexone and its major metabolite, 6β-naltrexol, was approximately 127 mL/min and 283 mL/min, respectively. However, the total systemic clearance of naltrexone was approximately 94 L/h in adults [40].

Naltrexone inhibits the metabolic activity of the enzymes CYP1A2, 2C9, 2D6, and 3A4. Therefore, it may readily interact with other drugs metabolized by these isoenzymes, thereby causing potential toxicity problems with these drugs [41].

### 4.2. Low Doses of Naltrexone

NTX in standard doses (25–50 mg) exhibits classical antagonism towards opioid receptors according to the dose–effect relationship. Recent literature reports that NTX is subject to hormetic mechanisms, and its pharmacodynamic effects depend on the concentration used [42]. On this basis, it was hypothesized that the effects of NTX are dose-dependent, but a research team led by McLaughlin proved that the duration of exposure of opioid receptor antagonists to the OGFr is a key factor on which the fate of cell divisions depends [43]. Low-dose naltrexone (LDN), ranging from 1 to 5 mg, causes a transient blockade of opioid receptors, which results in the “up-regulation” of the endogenous opioid system [44] (Figure 2). LDN exhibits strong, transient blocking action on the OGFr [45]. In experimental models, LDN competes with the OGF for binding sites, and after a short period of blockade, through feedback, promotes the release of endogenous opioid peptides as a result of compensatory up-regulation of the OGF–OGFr axis. Subsequently, the use of LDN leads to an increase in the expression of µ, δ, and OGFr opioid receptors [46,47,48], resulting in DNA replication inhibition, and thus limiting the proliferation of cancer cells [49]. NTX used in standard and higher doses showed a continuous antagonistic effect on the OGFr, thereby intensifying cell divisions [50].

### 4.3. The Impact of LDN on Healthy Tissues

A study conducted on tissues isolated from the brains of rats showed that LDN has a direct impact on the modulation of proteins associated with apoptosis. Under physiological conditions, LDN led to a decrease in pro-apoptotic proteins associated with the intrinsic pathway, such as Bad and Bax, while the expression of anti-apoptotic Bcl-2 and Bcl-xL, as well as effector caspase 3, remained unchanged. Moreover, LDN reduces the expression of FasL and Fas proteins in the cerebral cortex, which are also pro-apoptotic factors. The effects of LDN on healthy brain tissues suggest that endogenous opioid peptides play a role in inhibiting the entry of physiological cells into the apoptosis pathway, which in this case demonstrated that LDN may have a neuroprotective action [51].

### 4.4. The Impact of LDN on the Immune System

LDN is effective in modulating innate and acquired immune responses [52,53]. LDN can increase the phagocytic ability of macrophages, induce enhanced interactions between CD4+ T lymphocytes and macrophages, and stimulate the cytotoxic activity of NK cells [54]. LDN, applied in animal experimental models, increased the percentage of CD8+ T lymphocytes, which, along with NK cells, dominate the immune response directed against cancer cells through the synthesis of pro-inflammatory cytokines [55]. LDN enhances the production of immunoglobulin G2a (IgG2a) and interferon-γ (IFN-γ), which are responsible for the enhanced proliferation of Th1 lymphocytes. Exposure of lymphocytes to LDN results in an increased synthesis of pro-inflammatory cytokines, such as interleukins-2 (IL-2), interleukins-4 (IL-4), interleukins-6 (IL-6), and IFN-γ, by these cells [56]. Zijain and colleagues demonstrated that LDN induces the transition of macrophages from the M2 to the M1 type, which triggers the mobilization of these cells to secrete higher concentrations of tumor necrosis factor α (TNF-α), interleukin-12 (IL-12), and IL-6, as well as lower concentrations of interleukin-10 (IL-10) [57]. This mechanism is extremely important from an oncology perspective, as M2 macrophages significantly secrete interleukin-10 (IL-10), which promotes carcinogenesis in abnormal cells. The induction of M2 to M1 macrophage transition by LDN leads to reduced levels of IL-10. Moreover, M1 macrophages secrete cytokines that promote the release of reactive oxygen species and NO, having a direct cytotoxic effect on cancer cells. There is evidence in scientific studies that an increased number of M1 macrophages is associated with better prognoses for cancer patients [58,59,60]. The transient blockade of OGFr by LDN enhances the activity of endogenous opioid peptides, promoting the proliferation of B lymphocytes [61]. LDN mobilizes immune cells to act against pathogens in the early stages of the disease, suggesting its potential effectiveness in cancers caused by oncogenic viruses, such as cervical cancer associated with human papillomavirus (HPV) infection [62].

LDN also exhibits an immunomodulatory effect through antagonistic activity against Toll-like receptor-4 (TLR-4). TLR-4 is expressed on immunocompetent cells, such as macrophages and microglia, endothelial cells, as well as cancer cells, and it is a key point of attachment in the pathways triggering the inflammatory response of these cells [63]. TLR-4 detects numerous molecular patterns associated with pathogens and microorganisms, xenobiotics, and cell damage [64]. It triggers the primary response of myeloid differentiation primary response 88 (MyD88) and signaling dependent on the adaptor-inducing interferon-β (TRIF), which contains the TIR domain [65]. The adaptor proteins MyD88 and TRIF lead to the activation of NF-κB, which in turn leads to the upregulation of the release of pro-inflammatory cytokines, such as IL-1, interferon-β (INF-β), TNF-α, and NO [66]. LDN disrupts the TRIF signaling cascade, consequently reducing the synthesis of TNF-α and INF-β. Microglial cells expressing TLR-4 exposed to LDN ultimately show a weakened pro-inflammatory profile [67] (Figure 3).

In recent years, there has been a significant increase in the off-label use of LDN in the therapy of numerous autoimmune diseases, as well as non-cancer and neuropathic pain.

LDN administered to patients diagnosed with multiple sclerosis slowed the progression of the disease and prevented its relapses [68]. It was shown to inhibit the apoptosis of oligodendrocytes by reducing the activity of NO synthase, thereby inhibiting the synthesis of peroxynitrites, which in turn prevents the inhibition of glutamate transporters. As a result, the neurotoxicity of glutamate towards neurons and oligodendrocytes is reduced [69].

LDN therapy is also used in the treatment of fibromyalgia. Demonstrating LDN’s inhibitory effect on the activity of pro-inflammatory cytokines was the basis for conducting clinical trials using naltrexone at a dose of 4.5 mg among a group of patients suffering from fibromyalgia. It was shown that the use of LDN reduced the symptoms of fibromyalgia among 30.2% of study participants, compared to the group that received a placebo [70].

Among a group of 360 patients with rheumatoid arthritis who consistently used LDN, a clinically significant improvement in health status and a reduction in pain symptoms were noted, which translated into a reduction in their use of painkillers and anti-inflammatory drugs. However, there is an urgent need for randomized clinical trials to demonstrate the therapeutic efficacy of LDN in rheumatoid conditions [71].

### 4.5. The Impact of LDN on Cancer Cells

The effects of LDN on the endogenous opioid system suggest that this therapy could be applicable in cancer treatment, especially through the use of LDN to stimulate the OGF–OGFr axis after periodic receptor blockade. Additionally, LDN’s impact on modulating the immune response and endothelial cell angiogenesis seems to have a promising effect on limiting cancer cell invasion [29].

Different observations were made regarding the exposure of cancer cells to LDN. Colorectal cancer (CRC) cells exposed to LDN showed increased expression of proapoptotic factors—Bax, caspase-9, and caspase-3—and decreased expression of antiapoptotic proteins Bcl-2 and survivin [72]. The presented evidence suggests that the intrinsic pathway of apoptosis may play a significant role in inducing programmed cell death through LDN, and its effects are different in healthy and cancerous cells.

### 4.6. Differences in the Action of LDN and NTX at Standard Doses

Despite the increasing evidence of LDN’s unique role in inhibiting cancer progression, the precise mechanism of its action has not yet been fully elucidated. Liu et al. conducted studies on HCT116 human colorectal cancer cell lines to verify how cells respond to LDN and NTX at therapeutic doses. The authors demonstrated that, unlike NTX, LDN significantly affected the expression of genes associated with the regulation of the cell cycle, apoptosis, and autophagy. After exposing HCT116 cells to LDN, the expression of proapoptotic genes BAK1 and BAX was increased, while the expression of genes encoding cyclin B1 and cyclin-dependent kinases 1,4, and 6 (CDK1, CDK4, CDK6), which are involved in initiating DNA replication and further cell divisions, was decreased. Similar changes were not observed after exposing cells to NTX [73].

The mechanism of action of LDN and NTX at standard doses is presented in Figure 4 and Figure 5.

### 4.7. LDN in In Vitro and In Vivo Experimental Models

To verify the impact of LDN on OGFr in cervical cancer cells, in vitro studies were conducted on human cervical cancer cell lines Hela, Siha, C33A, and Caski. It was found that LDN causes an increase in the expression of OGFr in the Hela and Siha lines. Based on the results of the CCK-8 test, it was concluded that LDN can inhibit cancer cell proliferation in a time- and dose-dependent manner. The effect of LDN on the regulatory abilities of migration and invasion of the aforementioned cell lines was then examined, demonstrating that LDN treatment could significantly reduce the migration of Hela cells [74].

Ning et al. showed that LDN acts through the PI3K/AKT/mTOR pathway in Hela and Siha cervical cancer cells. The experiment showed that LDN significantly reduces the expression of PI3K, PDK1, and mTOR proteins; however, these changes were not significant for VEGF and AKT proteins [73].

Mingxing et al. evaluated the in vitro effect of LDN on the proliferation of human colorectal cancer cell lines SW480 and HCT116. The studies showed that LDN selectively inhibited the growth of the cells studied, and a clear decrease in cell colony count was observed in trials where higher concentrations of LDN were used, with a concentration of 1 mg/mL being classified as the most effective. The rate of colony formation by cells treated with 1 mg/mL LDN was significantly delayed, compared to the control group. Moreover, the researchers demonstrated that LDN induced apoptosis in the material studied [72].

The assessment of the impact of low-dose MNTX on the in vitro culture of human non-small cell lung cancer cells showed that the transient blockade of the OGFr by the treatment led to the inhibition of the growth and proliferation of cancer cells [10].

Studies of LDN in an in vivo mouse model with human cervical cancer xenografts revealed that LDN treatment not only directly blocked the epithelial-mesenchymal transition of cancer cells but also reduced the number of M2 macrophages directly associated with the tumor, resulting in a decrease in IL-10 synthesis [75].

Mingxing et al. investigated the impact of LDN on the progression of human colorectal cancer xenografts derived from SW480 and HTC116 cell lines in an in vivo model. The authors demonstrated that the expression of F4/80 and CD68 macrophages was significantly increased in the Foxn1nu (nude) mice group exposed to LDN, compared to the control group. Additionally, the concentrations of M1 macrophage phenotypic markers and TNF-α were higher than in the control group. LDN further increased the expression of OGFr, proapoptotic factors associated with Bax, caspase-9, caspase-3, and PARP, while reducing the expression of antiapoptotic Bcl-2, survivin, and Ki67. Their studies proved that exposure of human colorectal cancer cells to LDN reduces tumor size by intensifying the proliferation of M1 macrophages and directing cells towards the apoptosis pathway [72].

Studies conducted on Foxn1nu (nude) mice with SCC-1 oral squamous cell carcinoma cell line xenografts showed that LDN given to animals contributed to a significant reduction in tumor volume and mass, and DNA synthesis and cell divisions were significantly lowered [76].

In another study, Zagon et al. demonstrated that exposure of mice with immature neuroblastoma S20Y xenografts to LDN at a dose of 0.1 mg/kg b.w. also caused inhibition of tumor progression and invasion of the tumor into adjacent tissues [49].

Studies were also conducted evaluating the impact of LDN on human ovarian cancer SKOV-3 cells transplanted to male rats. After 40 days of treatment, a significant reduction in the number and mass of tumor nodules was observed compared to the control group, and the therapeutic effects of LDN were associated with inhibiting cancer cell proliferation and angiogenesis [45]. A comprehensive summary of data regarding the impact of LDN on selected types of cancer is presented in Table 1.

According to the authors, the presented results from the in vitro and in vivo models suggest that LDN has a high anticancer potential, and that its mechanism of action is pleiotropic. There are studies describing the use of OGF as a potential anticancer therapy [77,78,79]. The fact that the therapeutic effects of LDN in the context of inhibiting carcinogenesis are primarily associated with the transient inhibition of the OGF–OGFr pathway and a compensatory increase in OGF concentration further confirms the need for more research on the use of LDN in the context of treating oncological diseases.

### 4.8. Synergistic Therapy

The resistance of cancer cells to cytostatic drugs is a commonly observed occurrence in clinical practice and constitutes a significant problem in the effective therapy of neoplastic changes. Besides the lack of efficacy of the treatment applied, another barrier is the adverse effects, including direct cytotoxicity towards healthy tissues, which often translates into a deterioration of patients’ prognosis. A counterbalance to this phenomenon is the use of synergistic therapy, which is based on the utilization of two or more drugs that interact with each other based on additive synergy (the drugs have the same mechanism of action or a common target) and hyperadditive synergy (the drugs have different mechanisms of action or different targets, which makes the combined use of drugs more effective than the application of each one separately). Synergistic therapy allows for the optimization of treatment efficiency, overcoming the resistance of cancer cells, and reducing adverse effects [80].

Current evidence on the effect of LDN on inhibiting cancer progression remains largely unclear. The literature reports that LDN does not exert a direct antiproliferative effect on cancer cells, making it suitable for use in polytherapy with anticancer drugs, where their combination could promote higher efficacy of the treatment [77]. Moreover, the simultaneous use of cytostatic drugs in combination with LDN could allow for the determination of chemotherapy administration schemes that could translate into therapeutic synergy, reducing potential side effects and benefiting the patient. There are also scientific reports on the effectiveness of polytherapy involving LDN and drugs that do not show direct cytotoxic action on cancer cells, yet have shown effectiveness in inhibiting the progression of cancerous changes. The prospects of using LDN in combination with cytostatic drugs based on synergy interactions appear to be promising, as illustrated by the evidence presented in the following section of this review.

#### 4.8.1. LDN and Immunotherapy

IL-2 is an extremely important cytokine from the perspective of cancer immunotherapy, due to its pleiotropic action on the immune system [81]. IL-2 is primarily produced by CD4+ T lymphocytes, as well as by CD8+ T cells, NK cells, and activated dendritic cells (DC) [82,83]. IL-2 plays a key role in the differentiation of CD4+ T lymphocytes, promoting the cytotoxic activity of NK and CD8+ T cells, while simultaneously inhibiting the differentiation of Th17 lymphocytes [84,85]. In 1992, the FDA approved IL-2 for the treatment of metastatic renal cell carcinoma and in 1998 for therapy of metastatic melanoma. However, its use is limited due to the high risk of serious side effects, and it is not a first-line treatment [86]. Given the limitations of cancer monotherapy using IL-2, studies have been conducted on the use of IL-2 in combined immunotherapies with cytokines such as IFN-α [87,88,89] with LAK cells [90] and T cells [91], classical anticancer drugs [92,93] and immune checkpoint inhibitors [94,95]. Evidence describing the immunomodulating effects of LDN [96] and its impact on the synthesis and modulation of signaling pathways associated with IL-2 [19] supports the hypothesis that IL-2+LDN immunotherapy may provide therapeutic benefits in cancers showing increased expression of OGF and OGFr [96]. In a phase II clinical trial evaluating the effectiveness of IL-2+LDN immunotherapy, it was shown that blockade of the OGF–OGFr axis (which plays a physiological immunosuppressive role) by LDN effectively increased the anticancer activity of IL-2 in humans [97]. Lissoni et al. demonstrated that administering IL-2+LDN+melatonin (MLT) therapy to patients with solid tumors resulted in a significant increase in the average number of lymphocytes, compared to groups treated with only IL-2+MLT [98]. However, these data are limited, and there is a need for more research in this area.

The effects of LDN on the endogenous opioid system suggest that this therapy could be applicable in cancer treatment, especially the use of LDN to stimulate the OGF–OGFr axis after periodic receptor blockade. Additionally, LDN’s impact on modulating the immune response and endothelial cell angiogenesis seems to have a promising effect on limiting cancer cell invasion [29].

#### 4.8.2. LDN and Cisplatin

Studies conducted on mice with human ovarian cancer xenografts treated with LDN and cisplatin, as well as LDN and taxol, showed that the polytherapy regimen of LDN with cisplatin was more effective than the use of each of these drugs separately, as well as in combination with LDN with taxol. The assessment of apoptosis in SKOV-2 human ovarian cancer cell lines using the TUNEL assay revealed that groups exposed to LDN in combination with cisplatin or taxol had approximately a threefold higher percentage of apoptotic cells compared to the control group, which was only administered saline. DNA synthesis level assessments in cancer cells showed similarities in groups treated with LDN alone or LDN in combination with cisplatin. The density of blood vessels in the tumor microenvironment was reduced by 42–44% in the mouse groups, where LDN polytherapy with cisplatin was applied, compared to animals treated exclusively with LDN or cisplatin. Analyses of the results provide evidence of the positive therapeutic effect of the selected polytherapy, as LDN significantly mitigated the adverse effects resulting from the direct cytotoxic action of cisplatin. Moreover, LDN increased the expression of OGF and OGFr, proving that LDN stimulates the endogenous opioid system, which inhibits the proliferation of cancer cells [96]. Western Blot analysis of OGFr expression showed an 87% increase in the expression of this receptor among mice treated with LDN, compared to the control group [44].

#### 4.8.3. LDN and Carboplatin

To verify the effectiveness of polytherapy involving LDN in combination with carboplatin in malignant breast tumors, studies were conducted on 60 female dogs of various breeds, diagnosed with malignant breast tumors. The results showed that the percentage of CD8+ lymphocytes and the concentrations of beta-endorphin and enkephalin were higher in animals treated with LDN with carboplatin compared to groups treated only with carboplatin or the control group. Hematological complications resulting from carboplatin action, such as leukopenia and anemia, were lower in groups where polytherapy involving LDN was used. Moreover, the survival rate and quality of life improvement were significantly greater among patients treated with LDN with carboplatin [53].

#### 4.8.4. LDN and 5-Fluorouracil

Aboalsoud et al. conducted preclinical studies on female mice subcutaneously implanted with EAC solid ovarian cancer cell lines to demonstrate the mechanism of action and effectiveness of therapy with 5-fluorouracil (5FU) (20 mg/kg b.w.) in combination with LDN (0.1 mg/kg b.w.). An increase in OGFr expression in cancer cells was observed. The collected data revealed that the time-dependent increase in volume and mass of the cultivated tumors was statistically significantly smaller after using polytherapy with 5FU and LDN compared to groups treated with 5FU or LDN alone. Histopathological analyses of tumor tissue sections from mice subjected to polytherapy with 5FU and LDN showed necrosis, vacuolar degeneration of cancer cells, and lymphocyte infiltration around the tumor. Immunohistochemical confirmation showed an increase in the expression of proapoptotic proteins p21 and p51 in the group of mice treated with 5FU and LDN compared to the control group. Staining for the antiapoptotic protein Bcl-2 revealed significantly lower expression compared to the control group. Flow cytometry analysis of the EAC cell population showed a significant increase in apoptotic cells in the group treated with 5FU + LDN compared to groups exposed only to LDN, 5FU, and the control [99].

#### 4.8.5. Low-Dose Methylnaltrexone and 5-Fluorouracil

Methylnaltrexone [MNTX], a methylated derivative of NTX, is a peripheral antagonist of opioid receptors. Like LDN, low-dose MNTX showed a synergistic effect with 5FU in inhibiting cancer progression. In vitro studies conducted on human colorectal cancer cell lines SW-480, human breast cancer MCF-7, and non-small cell lung cancer cells showed that low-dose MNTX with 5FU inhibited growth and proliferation by 63.5% in SW-480 cells, 58.3% in MCF-7 cells, and 81.3% in non-small cell lung cancer cells, compared to groups treated only with 5FU. Moreover, the percentage of cells in the G1 phase was higher after MNTX treatment. The study proves that using low-dose MNTX to treat constipation in cancer patients may have an additive effect, increasing the effectiveness of 5FU therapy [100].

In another study, authors presented evidence that low-dose MNTX in combination with 5FU and bevacizumab effectively inhibited VEGF-induced angiogenesis in human pulmonary microvascular endothelial cells, which are associated with tumor angiogenesis processes [101].

#### 4.8.6. Low-Dose Methylnaltrexone and Docetaxel

Masami et al. showed that the use of low-dose MNTX as an adjuvant in docetaxel therapy increases the effectiveness of the therapy by reducing the resistance of gastric cancer cells to docetaxel. Researchers performed xenografts on nude mice with 60As6 gastric cancer cell lines resistant to docetaxel treatment. The study showed that low-dose MNTX, by transiently blocking OGFr, inhibited the OGF-induced suppression of cancer cell growth, thereby significantly improving survival and treatment outcomes in the studied animals [102].

#### 4.8.7. LDN and Cannabidiol

In a study conducted by Massi et al., it was shown that cannabidiol (CBD), like LDN, can modulate the enzymatic pathways of proteins associated with apoptosis processes [103]. Based on this, Liu et al. conducted studies on human lung cancer cell lines A549, human colorectal cancer HCT116, and in vivo models with xenografts derived from these cell lines in mice, aiming to determine whether polytherapy using LDN and CBD could be applicable in the pharmacotherapy of these cancers. It was shown that combined treatment with LDN and CBD did not have a significant impact on the number of proliferating cells; however, sequential exposure of cells first to LDN and then to CBD had a significant effect on reducing the number of live cells and colony count. Additionally, sequential therapy with LDN and CBD caused a significant increase in cell sensitivity to oxaliplatin applied in suboptimal doses. Similarly, the mass and volume of tumors in mice were reduced after applying the described therapy scheme [104].

#### 4.8.8. LDN and Propranolol

Studies conducted on human breast cancer cell lines MDA-MB-231, MDA-MB-468, and T47D in vitro and in vivo models showed that LDN in combination with propranolol significantly inhibited cell proliferation. Moreover, the applied polytherapy affected the arrest of these cells in the G2/M phase, and the levels of expression of proapoptotic effector proteins Bax, p-Bax, caspase-3, CC3, and cytochrome c were elevated compared to groups treated with LDN monotherapy, propranolol, and the control group. The analysis of tumor sizes derived from xenografts in mice showed a significant reduction in the mass and volume of the cultivated lesions [105]. The results provide evidence of the synergistic effect of therapy with LDN and propranolol [106].

#### 4.8.9. LDN and Vitamin D

The use of polytherapy with LDN and vitamin D brought positive effects for a 58-year-old patient suffering from tonsillar-cystic tongue cancer without metastases. The patient refused conventional chemotherapy, radiotherapy, and surgical treatment, so the treating physician prescribed him therapy with LDN at an initial dose of 3 mg and vitamin D at a dose of 10,000 IU daily. Due to the good tolerance of LDN, the dose was increased to 4 mg, and vitamin C was also added at a dose of 2000 mg/day. The patient experienced significant improvement after 3 months of therapy, and assessments of the size and degree of tumor development two years after the start of treatment showed that the sizes of the tumor lesions had reduced from 3 cm to 1.6 cm, and the radiologist noted progressive tumor regression [107].

#### 4.8.10. LDN and α-Lipoic Acid

A clinical case was described in which a 64-year-old patient with metastatic renal cell carcinoma (RCC) was treated with combined therapy of LDN and α-lipoic acid (ALA). The effectiveness of the therapy was confirmed by normal glucose uptake in the left lung, to which the cancer had previously metastasized. Moreover, the patient reported less shortness of breath, improved well-being allowing a return to work, and a return to normal weight. The authors of the study hypothesized that LDN along with ALA contribute to the reduction of tumor mass and transition of the cancer into a dormant state. Positive effects of LDN+ALA therapy were also observed in a patient with pancreatic cancer with liver metastases. After long-term polytherapy, the patient reported significant improvement in quality of life, and the disappearance of cancer-related symptoms, which enabled him to return to work. Similar effects were noted in three additional patients diagnosed with pancreatic adenocarcinoma with liver metastases, B-cell lymphoma, and prostate adenocarcinoma [108].

The summary of the collected data regarding the use of LDN in polytherapy is presented in Table 2.

According to the authors, the multifaceted mechanism of action of LDN may be an excellent complement to chemotherapy, as it shows synergy with the presented cytostatics, while itself having no direct cytotoxic effect on healthy cells. The mechanism of action of drugs such as cisplatin and carboplatin is based on creating cross-links within DNA strands and between adjacent DNA strands in cancer cells, which prevents DNA replication and cell division. Additionally, platinum complexes also affect numerous metabolic functions of cells, directing them towards the apoptosis pathway [109]. 5-FU is an inhibitor of thymidylate synthase, which leads to a reduction in the concentration of thymidine monophosphate (TMP). A low level of TMP is associated with disruption of DNA replication and inhibition of cancer cell proliferation. Furthermore, 5-FU incorporates into DNA and RNA, disrupting their structure [110]. Docetaxel stimulates the formation of microtubules and the creation of abnormal configurations during mitotic divisions, preventing the separation of the mitotic spindle. It also inhibits the depolymerization of tubulin, leading to the accumulation of microtubule bundles in cells, which results in the cessation of their reorganization. Additionally, docetaxel can direct a cell towards the apoptosis pathway by increasing the regulation threshold of proteins p53 and p21 and decreasing the expression of Bcl-2 [111]. The use of LDN in combination with these drugs may enhance the antiproliferative effect by blocking the transition of cells into the G1/S phase, disrupting intracellular pathways associated with cell proliferation, and promoting intrinsic apoptosis through increased expression of proapoptotic proteins and executive caspases, while simultaneously reducing the expression of antiapoptotic proteins. Moreover, LDN stimulated NK function and INF-γ and IL-2 production.

### 4.9. LDN in Clinical Trials

Current data on the effectiveness of using opioid receptor antagonists in cancer treatment in clinical settings are limited.

A randomized, double-blind clinical trial was conducted in a group of 54 patients with malignant glioma. Patients received daily LDN at a dose of 4.5 mg for 16 weeks. The analyses of the collected results showed that LDN did not improve the quality of life in the study group compared to the placebo [112]. A clinical trial conducted on 21 patients with malignant glioma undergoing radiotherapy and NTX at a dose of 100 mg every other day showed that a year after the end of the study, the tumor regression rate in patients after radiotherapy combined with NTX was not statistically significantly higher than that in the group treated with radiotherapy alone. However, the survival rate one year after the end of therapy was significantly higher in patients who received NTX and radiotherapy [113].

A phase II clinical trial was undertaken to evaluate the effectiveness of LDN in the treatment of treatment-resistant metastatic melanoma, prostate cancer, and renal cancer. Patients were prescribed NTX at a dose of 5 mg/day in a 28-day cycle. The study was discontinued due to a lack of willing participants [114].

Currently, a phase I non-randomized clinical trial is underway to evaluate the effectiveness of polytherapy with LDN in combination with propranolol, ipilimumab, and nivolumab among 12 patients with advanced melanoma. The study aims to determine the efficacy and safety of using LDN in combination with propranolol in patients undergoing immunotherapy with ipilimumab and nivolumab. The estimated completion time for this phase of the study is September 2025 [115]. Data collected from clinical trials in which LDN was used in cancer therapy are presented in Table 3.

Janku et al. conducted a pooled analysis from two randomized clinical trials where low-dose MNTX therapy was used among cancer patients. The combined dataset from both studies includes a group of 363 patients, of which 229 patients had advanced cancer, and 134 patients had advanced stages of the disease, requiring the administration of opioid receptor antagonists to treat constipation caused by opioid analgesic therapy. These studies showed that the use of low-dose MNTX significantly impacted extending life and improving prognosis compared to the patient group receiving a placebo; however, these data are limited, and more clinical studies are necessary [116].

## 5. Summary of the LDN Effects on Cancer Cells

This review demonstrates that existing research on the application of LDN therapy, either alone or in combination with other cytostatics, focuses on a group of cancers that are predominantly of epithelial origin and with a malignant character. The common feature of these cancers is primarily the increased expression of OGFr, and the therapeutic effects of LDN are mainly due to its transient properties, inhibiting this receptor. Most studies are conducted on cell lines derived from human ovarian cancer, breast cancer, colorectal cancer, and colon cancer. Independent results allow us to conclude that the consequence of the interaction between LDN and the OGF–OGFr axis is primarily a change in cell signaling associated with pathways of proteins inhibiting the cell cycle (an increased expression of p16, p21, and p51, along with a decreased expression of CDK group proteins—CDK1, CDK2, and CDK4). The effect of LDN on apoptosis was also similar in the studied cells, primarily involving an increase in the expression of proteins Bax, Bad, Bik, PARP, and executioner caspases 3 and 6, as well as initiating caspase-9, and a decrease in the expression of Bcl-2 and survivin. The consequence of LDN’s antagonism towards the µ-opioid receptor is the direct inhibition of endorphin synthesis and the inhibition of EGFr, which leads to the blocking of the epithelial-mesenchymal transition of cancer cells. LDN also exhibits other pleiotropic effects, focusing on modulating the activity of the immune system through a compensatory increase in OGF synthesis and antagonism towards TLR-4 receptors. Particularly important is the promotion of NK cell activity and the increase in the synthesis of pro-inflammatory cytokines IL-2, IL-4, IL-6, IFN-γ, and TNF-α. Also significant is the induction of M2 macrophage transition to M1 caused by LDN, as this leads to a reduction in IL-10 levels promoting carcinogenesis in damaged cells. The results included in the review also indicate that animals treated with LDN had an induced immune response against cancer cells. The evidence collected in this review allows us to conclude that LDN possesses numerous positive effects in the context of cancer therapy and can be applied as an adjuvant in cytostatic treatment, due to its potential synergistic effects in combination with these drugs.

## 6. Further Perspectives

Numerous pieces of evidence from preclinical studies and described clinical case series, with the independent use of LDN or its application as an adjuvant to classical chemotherapy, or in combination with drugs without direct cytostatic action, indicate that low-dose naltrexone has significant anticancer potential. Particularly promising seems to be the sensitizing action of resistant cancer cells to the applied treatment, as such a strategy will allow not only for more effective pharmacotherapy but also for the use of lower doses of cytotoxic drugs, which in turn translates into a reduction of severe adverse effects of chemotherapy. However, this evidence is still too limited. Hence, there is a necessity to conduct more studies, especially clinical trials on large patient groups, to confirm the existing hypotheses regarding the positive effects of LDN in cancer therapy.

The evidence presented in this review demonstrates that the mechanism of action of LDN is pleiotropic. The consequences of the transient inhibition of the OGF–OGFr axis translate into positive effects in inhibiting the growth, proliferation, and survival of cancer cells, especially in epithelial-origin tumors characterized by increased OGFr expression. LDN may also be associated with promoting apoptosis through the intrinsic pathway by increasing the expression of proapoptotic proteins while simultaneously decreasing the expression of antiapoptotic proteins. Moreover, the immunomodulating properties and inhibition of blood vessel angiogenesis in the tumor microenvironment lay the groundwork for conducting more extensive research that could qualify LDN as an adjuvant in synergistic therapy. According to the authors, it is worthwhile to expand the area of research on LDN to other types of cancers, particularly those associated with the nervous systems and hematologic cancers, since preliminary evidence illustrating the multidirectional LDN action suggests the existence of numerous potential targets through which this drug could demonstrate significant therapeutic benefits. In addition, the number of studies on the use of LDN as an adjuvant in chemotherapy is still small, so there is a need to check whether LDN enters into synergistic interactions with other cytostatics, especially with drugs from the anthracycline antibiotics group and mitosis inhibitors, because, based on the current evidence, it is possible to hypothesize that the mechanism of action of LDN would be an excellent complement to these drugs, which could bring direct benefits to oncology patients.

## 7. Conclusions

The current state of knowledge regarding the use of LDN in cancer treatment indicates that this drug has high therapeutic potential, particularly as an adjuvant for both traditional chemotherapy and new treatment methods, such as immunotherapy. The multifaceted action of LDN, leading to the inhibition of cancer progression, represents a new perspective in the therapy of oncological diseases, whose incidence is steadily increasing despite numerous effective treatment methods. Despite existing evidence, there is still an urgent need for more research on LDN in cancer therapy, especially randomized clinical trials on large groups of patients, which would allow for the verification of the rationale for using LDN in the group of oncological diseases.

## Figures and Tables

**Figure 1 cancers-16-01240-f001:**
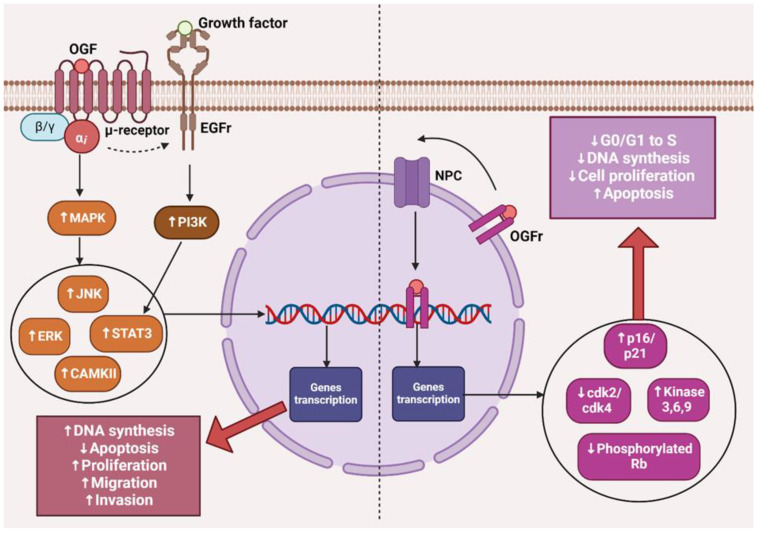
The effects of µ-opioid receptor and OGFr signaling in cancer cells. The image was created with BioRender. Abbreviations: OGF—opioid growth factor, OGFr—opioid growth factor receptor, EGFr—epidermal growth factor receptor, NPC—nuclear pore complex, MAPK—mitogen-activated protein kinases, PI3K—Phosphoinositide 3-kinase, JNK—c-Jun N-terminal kinases, ERK—extracellular signal-regulated kinases, STAT3—signal transducer and activator of transcription 3, CAMKII—calcium/calmodulin-stimulated protein kinase II, CDK2—cyclin-dependent kinase 2, CDK4—cyclin-dependent kinase 4, Rb—retinoblastoma protein.

**Figure 2 cancers-16-01240-f002:**
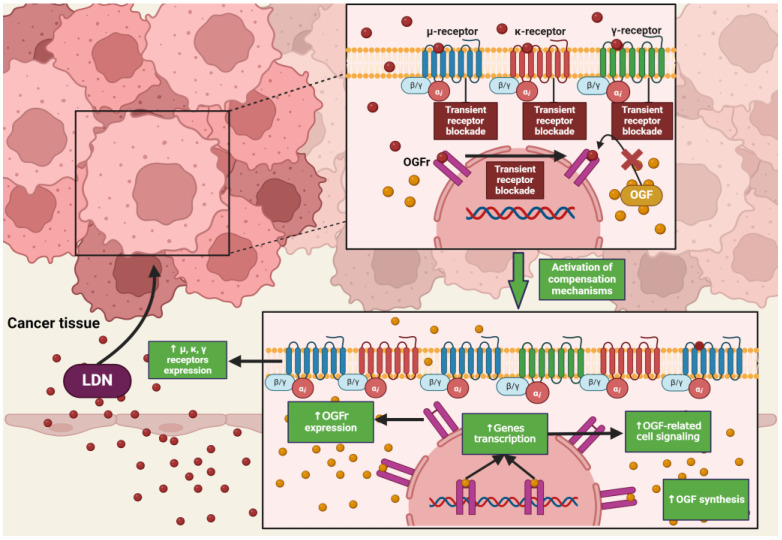
The impact of LDN on opioid receptor signaling in cancer cells. The image was created with BioRender. Abbreviations: OGF—opioid growth factor, OGFr—opioid growth factor receptor, LDN—low-dose naltrexone.

**Figure 3 cancers-16-01240-f003:**
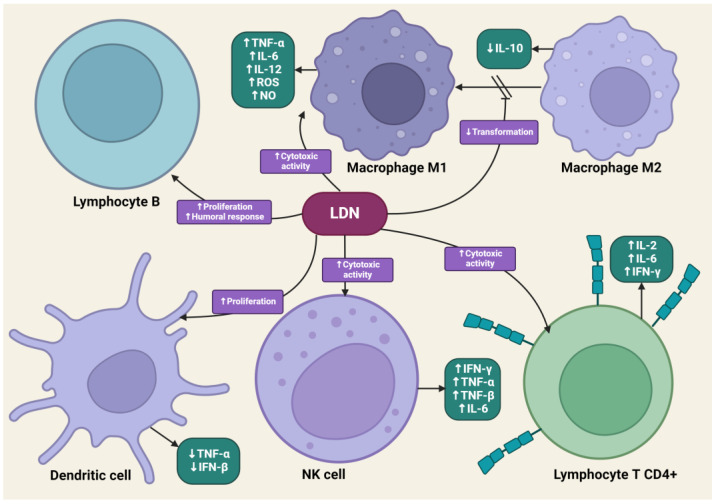
The impact of LDN on immune system cells. The image was created with BioRender. Abbreviations: CD4+—CD4-positive T lymphocyte, NK—natural killer cell, IL-2—interleukin-2, IL-6—interleukin-6, IL-10—interleukin-10, IL-12—interleukin-12, IFN-β—interferon beta, IFN-γ—interferon-gamma, LDN—low-dose naltrexone, NO—nitric oxide, ROS—reactive oxygen species, TNF-α—tumor necrosis factor-alpha, TNF-β—tumor necrosis factor beta.

**Figure 4 cancers-16-01240-f004:**
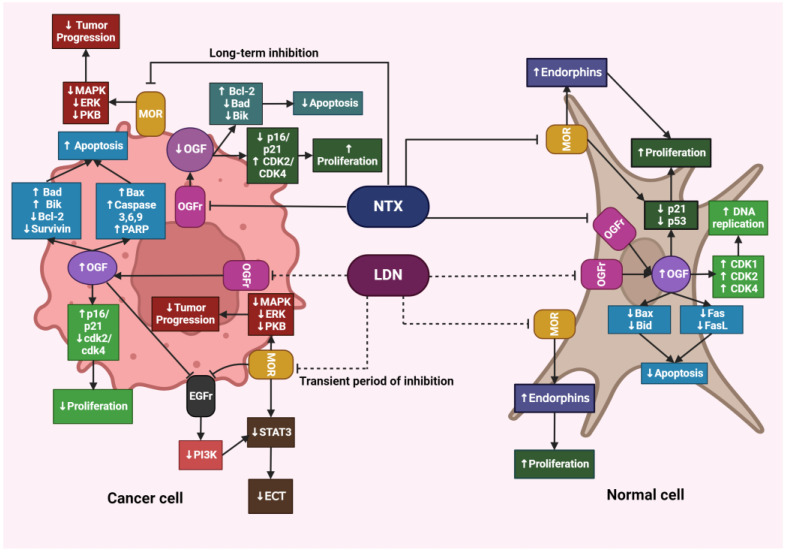
The effects of LDN and NTX at therapeutic doses on cancerous and healthy cells. The image was created with BioRender. Abbreviations: NTX—naltrexone, LDN—low-dose naltrexone, MOR—μ-opioid receptors, MAPK—mitogen-activated protein kinase, ERK—extracellular signal-regulated kinase, PKB—protein kinase B, OGF—opioid growth factor, OGFr—opioid growth factor receptor, Bcl-2—B-cell lymphoma protein, Bad—Bcl-2-associated agonist of cell death promoter, Bik—Bcl-2-interacting killer, Bax—Bcl-2-associated X protein, Bid—BH3-interacting domain death agonist, CDK1—cyclin-dependent kinase 1, CDK2—cyclin-dependent kinase 2, CDK4—cyclin-dependent kinase, FasL—Fas-ligand, PARP—poly (adenosine diphosphate [ADP]) ribose polymerase, EGFr—epidermal growth factor receptor, PI3K—phosphoinositide 3-kinase, STAT3—signal transducer and activator of transcription 3, ECT—epithelial-mesenchymal transition.

**Figure 5 cancers-16-01240-f005:**
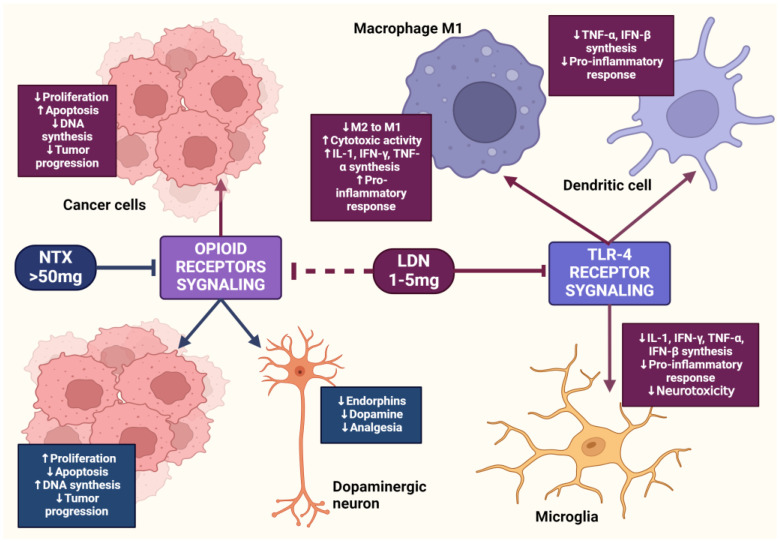
The mechanism of action of LDN and NTX at standard doses. The image was created with BioRender. Abbreviations: LDN—low-dose naltrexone, NTX—naltrexone, IL-1—interleukine 1, TNF-α—tumor necrosis factor-alpha, IFN-β—interferon beta, IFN-γ—interferon-gamma.

**Table 1 cancers-16-01240-t001:** The impact of LDN and low doses of MNTX on cancer cells in in vitro and in vivo models.

LDN/Low-Dose MNTX Concentration/Dose	Cancer	Results	References
LDN 0.5, 1.5, 2, 3 and 5 mg/mL	In vitro model of Hela and Siha, cervical cancer cells	LDN inhibits the proliferation of cervical cancer cells in a time- and dose-dependent manner. After 48 h of LDN treatment, the IC50 was 1.26 mg/mL. After treatment with LDN for 48 h, the inhibition rates of different concentrations (0.5 mg/mL, 1.5 mg/mL, 2 mg/mL, 3 mg/mL, 5 mg/mL) were 17.27 ± 5%, 47.44 ± 3%, 68.59 ± 4%, 84.68 ± 1%, and 95.47 ± 1%, respectively.	[73]
LDN 1 nM LDN 10 nM	In vitro model of human colorectal cancer cell lines HCT116 and human lung cancer cell lines A549	Cell counting experiments revealed that the reduction in cell number was associated with a fall in cell viability, which suggests an active cytotoxic response was achieved. Flow cytometric analysis of the cell cycle showed significant increases in the sub-G1 peak following an LDN-then-recovery schedule with concomitant emptying of cells from G1 and G2.	[72]
Low-dose MNTX 0.10–100 nM	In vitro model of human non-small cell lung cancer cells lines NSCLC	Treatment with MNTX inhibited cell invasion and anchorage-independent growth by 50–80%.	[10]
LDN intraperitoneal (IP) injection 0.1 mg/kg	SCC-1 oral squamous cell carcinoma xenografts in Foxn1nu (nude) mice	LDN increased the latency from visible to measurable tumors up to 1.6-fold. OGF, low-dose naltrexone, and imiquimod treatment markedly reduced tumor volume and weight and decreased DNA synthesis in tumors.	[76]
LDN 0.1 mg/kg	SKOV-3 human ovarian cancer xenografts in athymic nu/nu female mice	LDN-treated mice displayed a visible reduction in tumor burden relative to the control group. Compared to the total number of nodules detected in the control group, animals treated with LDN displayed a 39% reduction.	[45]
LDN 0.5 mg/kg, LDN 5 mg/kg LDN 10 mg/kg	Hela and Siha human cervical cancer xenografts in BALB/C nude mice	LDN significantly decreased the expression of PI3K, PDK1, and mTOR. There was no difference in the expression of VEGF and AKT, but the expression of pVEGFR2 and pAKT was downregulated. The expression of pVEGFR, PI3K, PDK1, pAKT, and mTOR significantly reduced after LDN treatment, especially in the 10 mg/kg group. Compared to the control group, the 10 mg/kg LDN treatment group showed significant differences in tumor growth inhibition from day 22 of the treatment, while the 5 mg/kg LDN-treated group showed such differences from day 31. The time of significant difference in mice treated with 0.5 mg/kg LDN was 34 days	[74]
LDN 0.5 mg/kg, LDN 5 mg/kg, LDN 10 mg/kg	Human cervical cancer cell lines Hela and Siha, xeno-grafts in BALB/C nude mice	The ratio of M2 macrophage membrane markers labeled with CD206+ showed a decrease in the LDN group compared with the control group. The proportion of TAMs significantly reduced after LDN treatment, especially in the 10 mg/kg group. LDN suppressed the M2 macrophages and reduced the expression of IL-10.	[73]

**Table 2 cancers-16-01240-t002:** The results of using LDN in polytherapy with cytostatics and other drugs with potential antitumor mechanisms.

Co-Treatments	Cancer	Mechanism/Results	References
LDN (10–5 mol/L), Taxol (10–9 or 10–10 mol/L), Cisplatin (0.01 or 0.001 µg/mL)	In vitro studies conducted on human ovarian cancer cell line SKOV-3	The number of cells exposed short-term to LDN and taxol was 36–61% lower compared to cells exposed only to LDN, and 19–31% lower compared to cells exposed only to taxol. The number of cells exposed to the short-term effects of LDN and cisplatin was reduced by 21–42% compared to cells exposed only to cisplatin.	[44]
Methylnaltrexone (1 µM) 5-Fluorouracil (10 µM)	In vitro studies conducted on human colorectal cancer cell lines SW-480, human breast cancer MCF-7, and non-small cell lung cancer cells	Inhibition of growth and proliferation by 63.5% in SW-480 cells, 58.3% in MCF-7 cells, and 81.3% in non-small cell lung cancer cells compared to groups treated only with 5FU.	[100]
MNTX (100 nmol/L), 5-FU (5 μmol/L), Bevacizumab (25 ng/mL)	In vitro studies conducted on human pulmonary microvascular EC (HPMVEC)	Methylnaltrexone (MNTX), synergistically with 5-FU and bevacizumab, inhibited vascular endothelial growth factor (VEGF)-induced human pulmonary microvascular endothelial cell (EC) proliferation and migration. MNTX inhibited EC proliferation with an IC(50) of approximately 100 nmol/L. The addition of MNTX to EC shifted the IC(50) of 5-FU from approximately 5 micromol/L to approximately 7 nmol/L. The addition of 50 MNTX shifted the IC(50) of bevacizumab in inhibiting EC migration from approximately 25 to approximately 6 ng/mL. RPTPμ activation inhibits VEGF-induced Src activation (target of bevacizumab). MNTX-induced Src inactivation results in activation of p190 RhoGAP and inhibition of active RhoA, which prevents reorganization of the actin cytoskeleton (targeted by 5-FU) and the resulting EC proliferation (targeted by 5-FU) and migration.	[101]
Naltrexone (10 nM–10 µM) Cannabidiol (CBD) (1 µM)	A549 (human lung cancer) and HCT116 (human colorectal cancer) cells	LDN and CBD reduced the number of cells. There was a 35% reduction in cell numbers when using LDN before CBD compared to a 22% reduction when using CBD before LDN.	[104]
LDN (0.1 mg/kg daily), Taxol (3 mg/kg, days 0, 7, 14, 21, 28, 35), Cisplatin (4 mg/kg days 0 and 7) intraperitoneal injections	Human ovarian cancer xenografts in female nude mice	Administration of NTX for six hours every two days, but not continuously, reduced DNA synthesis and cell replication compared to the control group. The combination of LDN with cisplatin, but not taxol, resulted in an additive inhibitory effect on tumorigenesis with enhanced depression of DNA synthesis and angiogenesis.	[44]
LDN (0.1 mg/kg), 5FU (20 mg/kg) subcutaneous injection	Human ovarian cancer xenografts in nude mice	A decrease in tumor mass and volume and an increase in the number of splenocytes, with a tendency to decrease the number of MDSC cells were observed. LDN led to an increase in OGFr both alone and in combination with 5FU, increased serum IFN-γ levels, but decreased when combined with 5-FU. The use of LDN and 5FU increased the expression of p21 and decreased Bcl2.	[99]
Low-Dose Methylnaltrexone (0.3 mg/kg) Docetaxel (Doc) (0.5 mg/kg)	60As6 human gastric cancer xenograft in female C.B17/Icr-scid mice	The growth of cells obtained from mice treated with a low-dose MNTX and Doc was significantly lower compared to mice treated with Doc only (Doc: 65.3 ± 6.6%, Doc/MNTX: 40.5 ± 7.1%). The use of Doc and low-dose MNTX polytherapy significantly extended life and alleviated cancer-related pain compared to mice treated with Doc only.	[102]
LDN (1.2 µg/mouse), CBD (35 µg/mouse), Gemcitabine (9 µg/mouse)	HCT116 colon cancer xenograft in athymic nu/nu BALB/c mice	The use of both compounds enhanced the effects of gemcitabine, without toxic effects.	[104]
NTX (0.001 µM to 200 µM) Propranolol (PRO) (0.001 µM to 200 µM)	Human breast cancer cells MDA-MB-231, MDA-MB-468, and T47D MDA-MB-231 xenograft in nude rat	Antitumor effects were observed due to the arrest of cell growth. NTX promoted PRO effects on expanded NK cells from the spleens and PBMCs of tumor xenografted animals. PRO and NTX increased the levels of NK cell-modulating cytokines while decreasing the levels of Th1 inflammatory cytokines.	[105]
LDN (0.1 mg/kg dose every 24 h for 24 weeks) orally, Carboplatin (300 mg/m^2^) intravenously	60 female dogs with mammary neoplasia	The higher serum concentrations of beta-endorphin and met-enkephalin, fewer chemotherapy-related side effects, and better quality of life and survival rates in the LDN-treated groups than in LDN-untreated groups. Evaluation of clinical and pathological parameters indicated a significant association between the use of LDN and prolonged survival, as well as enhanced quality of life.	[53]
NTX (100 mg) orally IL-2 (6 million lU/day subcutaneously for 6 days/week for 4 weeks)	14 consecutive untreatable metastatic solid tumor patients	The concomitant administration of NTX induced a significantly higher increase in lymphocyte mean number than that achieved with IL-2 plus MLT alone.	[44]
LDN (4.5 mg) α-Lipoic Acid (ALA) (300–600 mg)	64-year-old male patient diagnosed with metastatic renal cell carcinoma (RCC)	ALA could inhibit cancer cell growth by inhibiting the pro-inflammatory transcription factor, nuclear factor κ light chain enhancer of activated B cells (NF-κB). ALA, by inhibiting pyruvate dehydrogenase kinase (PDK), increases the activity of pyruvate dehydrogenase (PDHC), i.e., enzymes in the Warburg effect, inhibiting tumor development. Short-term opioid receptor blockade caused by LDN increases the production of enkephalin peptide, which, upon binding to OGFr, inhibits the proliferation of cancer cells.	[108]
LDN (3 mg) Vitamin D (10,000 IU daily)	58-year-old patient suffering from tonsillar-cystic tongue cancer without metastases	The patient has achieved nearly a four-year remission of his cancer based on his clinical status and the last MRI scan. LDN increases levels of the endogenous opioid methionine-enkephalin, which regulates cell proliferation and may inhibit the growth of cancer cells.	[107]

**Table 3 cancers-16-01240-t003:** A summary of existing clinical trials on the use of LDN in the treatment of cancer.

NCT Number	Status	Cancer	Treatment	Phase	Participants	Results/Comments	References
NCT05968690	Study Start (Actual)	Advanced Melanoma	Propranolol 30 mg + Naltrexone 4.5 mg	I	12	Study Completion (Estimated) 30 September 2025	[109]
NCT01650350	Enrollment (Actual)	Melanoma, Prostate Cancer, Renal Cancer	LDN, 5 mg/day − (1 cycle = 28 days)	II	7	Results N/A	[108]
NCT01303835	Enrollment (Actual)	Glioma	LDN, 4.5 mg	II	110	QOL and fatigue changes between baseline and post-concurrent chemotherapy and radiation therapy were not significantly different between patients receiving LDN or placebo.	[107]

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
