# Peer review of "Low-Dose Naltrexone as an Adjuvant in Combined Anticancer Therapy"

_cancers, 2024, doi:10.3390/cancers16061240_

Round 1

Reviewer 1 Report

Comments and Suggestions for Authors

This review deals with the effect of low dose naltrexone (LDN) and cancer. Actually, this review is well organized and written. The different effects of LDN have been described well, and several details on the mode of action of naltrexone help the reader to understand the main message of this manuscript. Also, the authors inserted information to distinguish the effect observed at low doses compared from those at high doses.

It is important that the authors show better in a figure the effects of LDN. Indeed, from the first figure, it is not clear that this treatment can lead to decrease of proliferation of cancer cells.

Actually, the authors stated that naltrexone is an antagonist of opioid receptors but to this reviewer this function is not shown in any figure and also the mechanism of action of LDN described on lines 117-132 is not shown.

I would strongly suggest a figure that explain better this point together with the effects of high doses.

Comments on the Quality of English Language

English is good.

Author Response

We would like to take this opportunity to deeply thank the Reviewer who identified the parts of our manuscript that required corrections or modifications. Please find the response to the Rewiever’s comments below.

REVIEWER #1:

  1. This review deals with the effect of low dose naltrexone (LDN) and cancer. Actually, this review is well organized and written. The different effects of LDN have been described well, and several details on the mode of action of naltrexone help the reader to understand the main message of this manuscript. Also, the authors inserted information to distinguish the effect observed at low doses compared from those at high doses.

We sincerely thank you for your feedback.

It is important that the authors show better in a figure the effects of LDN. Indeed, from the first figure, it is not clear that this treatment can lead to decrease of proliferation of cancer cells.

Thank you for this attention. Figure 1 shows the effects of μ-opioid receptor and OGFr signaling in cancer cells. Following the reviewer's suggestion, we have added Figure 2, which shows the effect of LDN on opioid receptor signaling in cancer cells. Moreover, the detailed mechanism leading to the antiproliferative effect in cancer cells is presented in Figure 3.

Actually, the authors stated that naltrexone is an antagonist of opioid receptors but to this reviewer this function is not shown in any figure and also the mechanism of action of LDN described on lines 117-132 is not shown.

Thank you for the suggestion. To better illustrate the mechanism of action of LDN in cancer cells, which was described in lines 117-132, a new Figure 2 has been prepared. Additionally, explanations of the abbreviations used in the figures have been placed under each figure to make it easier for readers to understand and comprehend the mechanisms presented.

Changes in the manuscript:

Page 5, section: “4. Naltrexone”, subsection: “4.2. Low Doses of Naltrexone”: A new figure No.2 presenting the impact of LDN on opioid receptor signaling in cancer cells  has been added.

Page 3, section: “The OGF-OGFr Axis”, Figure No. 1: Explanations of abbreviations have been added to the figure description.

Page 5, section “4.Naltrexone”, subsection:”4.2. Low Doses of Naltrexone”, Figure No.2:

Explanations of abbreviations have been added to the figure description.

Page 6, section: “4.Naltrexone”, subsection: “4.4. The impact of LDN on the Immune System”, Figure No.3: Explanations of abbreviations have been added to the figure description.

Page 8, section: “4.Naltrexone”, subsection: “4.6. Differences in the Action of LDN and NTX at Standard Doses”, Figure No.4: Explanations of abbreviations have been added to the figure description.

Page 9 section: “4.Naltrexone”, subsection: “4.6. Differences in the Action of LDN and NTX at Standard Doses”, Figure No.5: Explanations of abbreviations have been added to the figure description.

I would strongly suggest a figure that explain better this point together with the effects of high doses.

We are grateful for pointing it out. In accordance with the Reviewer's suggestion, Figure 5 has been added, comparing the effects of LDN and naltrexone at standard doses, which will complement the information presented in Figure 4.

Changes in the manuscript:

Page 9, section:”4.Naltrexone”, subsection:“4.6 Differences in the Action of LDN and NTX at Standard Doses”: A new figure presenting the mechanism of action of LDN and NTX at standard doses has been added.

Reviewer 2 Report

Comments and Suggestions for Authors

The submitted review articles describes the effect of low dose naltrexone on cancer cell survival, proliferation and invasion. The view addresses an interesting topic and may profit from some additions.

  1. The difference in the oral medication between standard and low dose are marked. What are the resulting blood levels, pharmacokinetics (inter-individual variations, impact of excretion and liver metabolism, CYP metabolization)?
  2. Which are the proposed mechanisms for the synergistic effects?
  3. An overview table with information about type of the study (in vitro, in vivo), cancer type, doses, etc. would be helpful.

Minor

Fig. 2: typo in “Limphocyte”

Fig. 2: add abbreviations in the legends

Author Response

We would like to take this opportunity to deeply thank the Reviewer who identified the parts of our manuscript that required corrections or modifications. Please find the response to the Rewiever’s comments below.

REVIEWER #2:

The submitted review articles describes the effect of low dose naltrexone on cancer cell survival, proliferation and invasion. The view addresses an interesting topic and may profit from some additions.

We sincerely thank You for your feedback.

The difference in the oral medication between standard and low dose are marked. What are the resulting blood levels, pharmacokinetics (inter-individual variations, impact of excretion and liver metabolism, CYP metabolization)?

Thank You for the suggestion. In accordance with the Reviewer's recommendation, a new subsection has been added that details the pharmacokinetics of naltrexone, noting that data regarding the pharmacokinetics of LDN are still limited.

Changes in the manuscript:

      Page 4, section “4. Naltrexone”: A new subsection “4.1. Naltrexone pharmacokinetics” has been added (lines132-164): Pharmacological data describing the safety profile of naltrexone reveal that its use at a dose of 300 mg daily may lead to liver cell damage [37]. However, naltrexone at a dose of 50–100 mg and lower is considered completely safe for humans. This is partly due to the poor bioavailability (5-40% due to the first-pass effect) of naltrexone after oral administration, which means that systemic side effects following this route of administration are minimal. In turn, parenteral administration of naltrexone may potentially lead to side effects [38]. Regarding LDN, data on the actual effects of the drug are still limited. Results from clinical trials indicate that all low-dose naltrexone, very low-dose naltrexone, and ultra-low-dose naltrexone are well tolerated, even with concurrent opioid therapy.

      After parenteral administration, naltrexone is rapidly distributed in the body, easily crosses the placenta and binds relatively poorly to albumin. Its metabolism takes place in the liver by dihydrodiol dehydrogenases into 6β-naltrexol (6β-hydroxynaltrexone) [39]. The biotransformation of naltrexone is individually variable in the liver in both children and adults and depends primarily on genetic variability, age and sex [40]. There are indications that the AKR1C4 genotype has a large impact on the biotransformation of naltrexone. In men, due to the high concentration of testosterone and dihydrotestosterone, the formation of 6βN is inhibited. Literature data indicate that adults treated with oral naltrexone had a greater than 10-fold variability in systemic exposure (e.g., Cmax and area under the curve). According to some au-thors, the average half-lives of naltrexone and 6β-naltrexol were approximately 4 and 12 hours, respectively [41]. According to others, the serum half-life of naltrexone in adults ranged from 30 to 81 minutes (mean 64 ± 12 minutes). In neonates, the mean plasma half-life was 3.1 ± 0.5 hours. Naltrexone administered orally or intravenously is approximately 25 to 40% renal excreted as metabolites within 6 hours, approximately 50% within 24 hours and 60 to 70% within 72 hours [40].

      The renal clearance of naltrexone and its major metabolite, 6β-naltrexol, was ap-proximately 127 ml/min and 283 ml/min, respectively. However, the total systemic clearance of naltrexone was approximately 94 l/h in adults [41].

      Naltrexone inhibits the metabolic activity of the enzymes CYP1A2, 2C9, 2D6 and 3A4, and therefore may readily interact with other drugs metabolized by these isoenzymes, thereby causing potential toxicity problems with these drugs [42].

Which are the proposed mechanisms for the synergistic effects?

We are grateful for poiting it out.  In line with the Reviewer's suggestion, the "Synergistic Therapy" section has been expanded to include proposed mechanisms of synergistic effects.

Changes in the manuscript:

Page 12, section “4.Naltrexone”, subsection “4.8. Synergistic Therapy”: A new paragraphs had been added (lines 368-379): The resistance of cancer cells to cytostatic drugs is a commonly observed occurrence in clinical practice and constitutes a significant problem in the effective therapy of neoplastic changes. Besides the lack of efficacy of the treatment applied, another barrier is the adverse effects, including the direct cytotoxicity towards healthy tissues, which often translates into a deterioration of patients' prognosis. A counterbalance to this phenomenon is the use of synergistic therapy, which is based on the utilization of two or more drugs that interact with each other based on additive synergy (the drugs have the same mechanism of action or a common target) and hyperadditive synergy (the drugs have different mechanisms of action or different targets, which makes the combined use of drugs more effective than the application of each one separately). Synergistic therapy allows for the optimization of treatment efficiency, overcoming the resistance of cancer cells, and reducing adverse effects [82].

Page 12, section “4.Naltrexone”, subsection “4.8. Synergistic Therapy”: A new sentence has been added (lines 389-391): The prospects of using LDN in combination with cytostatic drugs based on synergy interactions appear to be promising, as illustrated by the evidence presented in the following section of this review.

Page 19, section “4.Naltrexone”, subsection “4.8. Synergistic Therapy”: A new paragraph has been added in the end of the subsection (lines 543-564):  According to the authors, the multifaceted mechanism of action of LDN may be an excellent complement to chemotherapy, as it shows synergy with the presented cytostatics, while itself having no direct cytotoxic effect on healthy cells. The mechanism of action of drugs such as cisplatin and carboplatin is based on creating cross-links within DNA strands and between adjacent DNA strands in cancer cells, which pre-vents DNA replication and cell division. Additionally, platinum complexes also affect numerous metabolic functions of cells, directing them towards the apoptosis pathway [112]. 5-FU is an inhibitor of thymidylate synthase, which leads to a reduction in the concentration of thymidine monophosphate (TMP). A low level of TMP is associated with disruption of DNA replication and inhibition of cancer cell proliferation. Furthermore, 5-FU incorporates into DNA and RNA, disrupting their structure [113]. Docetaxel stimulates the formation of microtubules and the creation of abnormal con-figurations during mitotic divisions, preventing the separation of the mitotic spindle. It also inhibits the depolymerization of tubulin, leading to the accumulation of microtubule bundles in cells, which results in the cessation of their reorganization. Additionally, docetaxel can direct a cell towards the apoptosis pathway by increasing the regulation threshold of proteins p53 and p21 and decreasing the expression of Bcl-2 [114]. The use of LDN in combination with these drugs may enhance the antiproliferative effect by blocking the transition of cells into the G1/S phase, disrupting intracellular pathways associated with cell proliferation, and promoting intrinsic apoptosis through increased expression of proapoptotic proteins and executive caspases while simultaneously reducing the expression of antiapoptotic proteins. Moreover, LDN stimulated NK function and INF-γ and IL-2 production.

An overview table with information about type of the study (in vitro, in vivo), cancer type, doses, etc. would be helpful.

Thank You for the suggestion. A table summarizing the information contained in the subsection: “LDN in In Vitro and In Vivo Experimental Models” has been added.

Changes in manuscript:

Page 10, Section 4. “Naltrexone”, subsection “4.7. LDN in In Vitro and In Vivo Experimental Models” : A table “The impact of LDN and low doses of MNTX on cancer cells in vitro and in vivo models” has been added as Table No.1.

Minor

Fig. 2: typo in “Limphocyte”

Thank You. The typo has been corrected to “Lymphocyte”.

Changes in the manuscript:

 Page 6, section “4.Naltrexone”, subsection “4.4. The impact of LDN on the Immune System”, Figure 3.: Typo in “Limphocyte” has been corrected to “Lymphocyte”

Fig. 2: add abbreviations in the legends

Thank You for pointing it out. Authors have also decided to add explanations of abbreviations under the remaining figures to make them more accessible to future readers.

Changes in the manuscript:

Page 5, section: “4. Naltrexone”, subsection: “4.2. Low Doses of Naltrexone”: A new figure No.2 presenting the impact of LDN on opioid receptor signaling in cancer cells  has been added.

Page 3, section: “The OGF-OGFr Axis”, Figure No. 1: Explanations of abbreviations have been added to the figure description.

Page 5, section “4.Naltrexone”, subsection:”4.2. Low Doses of Naltrexone”, Figure No.2:

Explanations of abbreviations have been added to the figure description.

Page 6, section: “4.Naltrexone”, subsection: “4.4. The impact of LDN on the Immune System”, Figure No.3: Explanations of abbreviations have been added to the figure description.

Page 8, section: “4.Naltrexone”, subsection: “4.6. Differences in the Action of LDN and NTX at Standard Doses”, Figure No.4: Explanations of abbreviations have been added to the figure description.

Page 9 section: “4.Naltrexone”, subsection: “4.6. Differences in the Action of LDN and NTX at Standard Doses”, Figure No.5: Explanations of abbreviations have been added to the figure description.

Reviewer 3 Report

Comments and Suggestions for Authors

The manuscript by Ciwun et al. is a review article focusing on low-dose naltrexone as an adjuvant in combined anticancer therapy. Thus, this review represents an update of recent advances in the field that is timely, well written, well illustrated, and easy to follow. While the reviews on this topic have been published in 2022 or earlier, I would like to suggest the authors to clearly state the purpose of the current review at the end of the introduction. Chapter 5 in the manuscript is currently titled "Summary". In my opinion, the title "Further Perspectives" is more appropriate, and I would like to encourage the authors to expand it by describing the future development of an area in more detail.

Author Response

We would like to take this opportunity to deeply thank the Reviewer who identified the parts of our manuscript that required corrections or modifications. Please find the response to the Rewiever’s comments below.

REVIEWER #3:

The manuscript by Ciwun et al. is a review article focusing on low-dose naltrexone as an adjuvant in combined anticancer therapy. Thus, this review represents an update of recent advances in the field that is timely, well written, well illustrated, and easy to follow.

We sincerely thank you for your feedback.

While the reviews on this topic have been published in 2022 or earlier, I would like to suggest the authors to clearly state the purpose of the current review at the end of the introduction.

We are grateful for pointing it out. In accordance with the Reviewer's suggestion, the purpose of this review has been detailed at the end of the "Introduction" section.

Changes in manuscript

Page 1, section “1.Introduction”: A new paragraphs had been added in the end of this section (lines 43-54): In this literature review, we presented research on the use of naltrexone and methylnaltrexone (MNTX) in the context of enhancing anticancer effects. Both of these compounds belong to opioid receptor antagonists, but due to their chemical structure (quaternary amine), MNTX does not penetrate the blood barrier and therefore its action is limited to peripheral receptors.

Available literature data, supported by the results of in vitro and in vivo studies, indicate the potential use of LDN as an adjuvant in combined anticancer therapy. The mechanism of this beneficial effect is not clear. It is the result of the influence on the opioid growth factor receptor (OGFr) axis, which results in reduced cell replication and an increase in the cytolytic activity of NK cells as well as stimulation INF-γ and IL-2 production. The existence of other additional mechanisms of action cannot be ruled out, therefore it is necessary to thoroughly understand the biological effects of naltrexone.

Chapter 5 in the manuscript is currently titled "Summary". In my opinion, the title "Further Perspectives" is more appropriate, and I would like to encourage the authors to expand it by describing the future development of an area in more detail.

Thank You for your suggestion. Section No. 5 has been changed from "Summary" to "Further Perspectives" and has been extensively expanded with the authors' insights on potential future research opportunities regarding the use of LDN in the synergistic therapy of cancers.

Changes in manuscript

Page 21, section  “6. Summary” has been renamed to “6. Further Perspectives”.

Page 21, section “6. Further Perspectives”: A new paragraph has been added in the end of this section (lines 637-656): The evidence presented in this review demonstrates that the mechanism of action of LDN is pleiotropic. The consequences of transient inhibition of the OGF-OGFr axis translate into positive effects in inhibiting the growth, proliferation, and survival of cancer cells, especially in epithelial-origin tumors characterized by increased OGFr ex-pression. LDN may also be associated with promoting apoptosis through the intrinsic pathway by increasing the expression of proapoptotic proteins while simultaneously decreasing the expression of antiapoptotic proteins. Moreover, the immunomodulating properties and inhibition of blood vessel angiogenesis in the tumor microenvironment lay the groundwork for conducting more extensive research that could qualify LDN as an adjuvant in synergistic therapy. According to the authors, it is worthwhile to expand the area of research on LDN to other types of cancers, particularly those associated with the nervous systems and hematologic cancers, since preliminary evidence il-lustrating the multidirectional LDN action suggests the existence of numerous potential targets through which this drug could demonstrate significant therapeutic bene-fits. In addition, the number of studies on the use of LDN as an adjuvant in chemotherapy is still small, so there is a need to check whether LDN enters into synergistic inter-actions with other cytostatics, especially with drugs from the anthracycline antibiotics group and mitosis inhibitors, because based on the current evidence, it is possible to hypothesize that the mechanism of action of LDN would be an excellent complement to these drugs, which could bring direct benefits to oncology patients.

Reviewer 4 Report

Comments and Suggestions for Authors

General: There are already reviews regarding low-dose naltrexone (LDN) as a potential cancer treatment. In the introduction section, the authors need to show more effort to identify the existing information gap, followed by how this review can fill the gap and provide updated or complementary information, helping readers to keep up with the latest developments in the field.  

Specific: Each section provides a descriptive overview of the respective topic but lacks thorough discussion or expert insights (from the authors). For instance, Section 4.7, focusing on synergistic therapy, lacks specificity. Synergistic therapy is a loosely defined term. Define the term and use the definition to organize discussions. One of the questions poses to researchers is how to quantitatively characterize and compare different treatments that combine LDN with other drugs. In additional, the authors have missed the discussion of LDN and chemotherapy. Key information in subsections 4.7.3-4.7.10 could probably be presented in a table, improving readability. Last but not least, the authors could have have discussed how LDN impacts specific cancer types.

Comments on the Quality of English Language

Minor editing of English language required.

Author Response

We would like to take this opportunity to deeply thank the Reviewer who identified the parts of our manuscript that required corrections or modifications. Please find the response to the Rewiever’s comments below.

REVIEWER #4:

General: There are already reviews regarding low-dose naltrexone (LDN) as a potential cancer treatment. In the introduction section, the authors need to show more effort to identify the existing information gap, followed by how this review can fill the gap and provide updated or complementary information, helping readers to keep up with the latest developments in the field. 

We are grateful for pointing it out. In accordance with the Reviewer's suggestion, the purpose of this review has been detailed at the end of the "Introduction" section.

Changes in manuscript

Page 1, section “1.Introduction”: A new paragraphs had been added in the end of this section (lines 43-54): In this literature review, we presented research on the use of naltrexone and methylnaltrexone (MNTX) in the context of enhancing anticancer effects. Both of these compounds belong to opioid receptor antagonists, but due to their chemical structure (quaternary amine), MNTX does not penetrate the blood barrier and therefore its action is limited to peripheral receptors.

Available literature data, supported by the results of in vitro and in vivo studies, indicate the potential use of LDN as an adjuvant in combined anticancer therapy. The mechanism of this beneficial effect is not clear. It is the result of the influence on the opioid growth factor receptor (OGFr) axis, which results in reduced cell replication and an increase in the cytolytic activity of NK cells as well as stimulation INF-γ and IL-2 production. The existence of other additional mechanisms of action cannot be ruled out, therefore it is necessary to thoroughly understand the biological effects of naltrexone.

Specific: Each section provides a descriptive overview of the respective topic but lacks thorough discussion or expert insights (from the authors). For instance, Section 4.7, focusing on synergistic therapy, lacks specificity. Synergistic therapy is a loosely defined term. Define the term and use the definition to organize discussions. One of the questions poses to researchers is how to quantitatively characterize and compare different treatments that combine LDN with other drugs. In additional, the authors have missed the discussion of LDN and chemotherapy. Key information in subsections 4.7.3-4.7.10 could probably be presented in a table, improving readability. Last but not least, the authors could have have discussed how LDN impacts specific cancer types.

We are grateful for pointing it out. In accordance with the Reviewer's suggestion, In the review, the discussion on the addressed issue has been expanded, and the authors' observations have been included. The definition of synergistic therapy has been elaborated, and references suggested by the Reviewer have been added in subsequent parts of the section. The mechanism of synergistic effects of LDN and chemotherapy has been discussed in detail. Moreover, a table summarizing the current data on possible polytherapies using LDN has been prepared. At the end of the review, a new chapter focusing on summarizing the impact of LDN on specific types of cancers has also been added.

Changes in manuscript

Page 12, section “4.Naltrexone”, subsection “4.7. LDN in In Vitro and In Vivo Experimental Models”: A new paragraph has been added in the end of this subsection (lines 359-365): According to the authors, the presented results from in vitro and in vivo models suggest that LDN has a high anticancer potential, and its mechanism of action is pleiotropic. There are studies describing the use of OGF as a potential anticancer therapy [79, 80, 81]. The fact that the therapeutic effects of LDN in the context of inhibiting carcinogenesis are primarily associated with the transient inhibition of the OGF-OGFr pathway and a compensatory increase in OGF concentration further confirms the need for more research on the use of LDN in the context of treating oncological diseases.

Page 12, section “4.Naltrexone”, subsection “4.8. Synergistic Therapy”: A new paragraph has been added in the beginning of subsection (lines 368-379): The resistance of cancer cells to cytostatic drugs is a commonly observed occurrence in clinical practice and constitutes a significant problem in the effective therapy of neoplastic changes. Besides the lack of efficacy of the treatment applied, another barrier is the adverse effects, including the direct cytotoxicity towards healthy tissues, which often translates into a deterioration of patients' prognosis. A counterbalance to this phenomenon is the use of synergistic therapy, which is based on the utilization of two or more drugs that interact with each other based on additive synergy (the drugs have the same mechanism of action or a common target) and hyperadditive synergy (the drugs have different mechanisms of action or different targets, which makes the combined use of drugs more effective than the application of each one separately). Synergistic therapy allows for the optimization of treatment efficiency, overcoming the resistance of cancer cells, and reducing adverse effects [82].

Page 12, section “4.Naltrexone”, subsection “4.8. Synergistic Therapy”: A new sentence shas been added in the end of subsection (lines 389-391): The prospects of using LDN in combination with cytostatic drugs based on synergy interactions appear to be promising, as illustrated by the evidence presented in the following section of this review.

Page 12, section “4.Naltrexone”, subsection “4.8. Synergistic Therapy” subsubsection “4.8.2. LDN and Cisplatin”: A new sentences has been added (425-433 lines) : The assessment of apoptosis in SKOV-2 human ovarian cancer cell lines using the TUNEL assay revealed that groups exposed to LDN in combination with cisplatin or taxol had approximately a threefold higher percentage of apoptotic cells compared to the control group, which was only administered saline. DNA synthesis level assessments in cancer cells showed similarities in groups treated with LDN alone or LDN in combination with cisplatin. The density of blood vessels in the tumor microenvironment was reduced by 42-44% in the mouse groups where LDN polytherapy with cisplatin was applied compared to animals treated exclusively with LDN or cisplatin.

Page 12, section “4.Naltrexone”, subsection “4.8. Synergistic Therapy” subsubsection “4.8.2. LDN and Cisplatin”: A new sentence has been added in 437-439 lines: Western Blot analysis of OGFr expression showed an 87% increase in the expression of this receptor among mice treated with LDN compared to the control group [101].

Page 15, section “4.Naltrexone”, subsection “4.8. Synergistic Therapy”: A table summarizing the use of LDN in synergistic therapy has been added as Table No. 1.

Page 19, section “4. Naltrexone”, subsection “4.8. Synergistic Therapy”: A new paragraph has been added in the end of section (lines 543-565): According to the authors, the multifaceted mechanism of action of LDN may be an excellent complement to chemotherapy, as it shows synergy with the presented cytostatics, while itself having no direct cytotoxic effect on healthy cells. The mechanism of action of drugs such as cisplatin and carboplatin is based on creating cross-links within DNA strands and between adjacent DNA strands in cancer cells, which pre-vents DNA replication and cell division. Additionally, platinum complexes also affect numerous metabolic functions of cells, directing them towards the apoptosis pathway [112]. 5-FU is an inhibitor of thymidylate synthase, which leads to a reduction in the concentration of thymidine monophosphate (TMP). A low level of TMP is associated with disruption of DNA replication and inhibition of cancer cell proliferation. Furthermore, 5-FU incorporates into DNA and RNA, disrupting their structure [113]. Docetaxel stimulates the formation of microtubules and the creation of abnormal con-figurations during mitotic divisions, preventing the separation of the mitotic spindle. It also inhibits the depolymerization of tubulin, leading to the accumulation of microtu-bule bundles in cells, which results in the cessation of their reorganization. Additional-ly, docetaxel can direct a cell towards the apoptosis pathway by increasing the regula-tion threshold of proteins p53 and p21 and decreasing the expression of Bcl-2 [114]. The use of LDN in combination with these drugs may enhance the antiproliferative effect by blocking the transition of cells into the G1/S phase, disrupting intracellular pathways associated with cell proliferation, and promoting intrinsic apoptosis through increased expression of proapoptotic proteins and executive caspases while simultaneously reducing the expression of antiapoptotic proteins. Moreover, LDN stimulated NK function and INF-γ and IL-2 production.

Page 20: A new section No. 5 “Summary of the LDN effect on cancer cells” has been added (lines 598-625): This review demonstrates that existing research on the application of LDN therapy alone or in combination with other cytostatics focuses on a group of cancers, which are predominantly of epithelial origin with a malignant character. The common feature of these cancers is primarily the increased expression of OGFr, and the therapeutic effects of LDN are mainly due to its transient properties inhibiting this receptor. Most studies are conducted on cell lines derived from human ovarian cancer, breast cancer, colorectal cancer, and colon cancer. Independent results allow us to conclude that the consequence of the interaction between LDN and the OGF-OGFr axis is primarily a change in cell signaling associated with pathways of proteins inhibiting the cell cycle (in-creased expression of p16, p21, p51, and decreased expression of CDK group proteins - CDK1, CDK2, CDK4). The effect of LDN on apoptosis was also similar in the studied cells, primarily involving an increase in the expression of proteins Bax, Bad, Bik, PARP, and executioner caspases 3 and 6, as well as initiating caspase-9, and a decrease in the expression of Bcl-2 and survivin. The consequence of LDN's antagonism towards the µ-opioid receptor is the direct inhibition of endorphin synthesis and the inhibition of EGFr, which leads to the blocking of the epithelial-mesenchymal transition of cancer cells. LDN also exhibits other pleiotropic effects, focused on modulating the activity of the immune system through a compensatory increase in OGF synthesis and antagonism towards TLR-4 receptors. Particularly important is the promotion of NK cell activity, the increase in the synthesis of pro-inflammatory cytokines IL-2, IL-4, IL-6, IFN-γ, and TNF-α. Also significant is the induction of M2 macrophages transition to M1 caused by LDN, as this leads to a reduction in IL-10 levels promoting carcinogenesis in damaged cells. The results included in the review also indicated that animals treated with LDN had an induced immune response against cancer cells. The evidence collected in the review allows us to conclude that LDN possesses numerous positive effects in the context of cancer therapy and can be applied as an adjuvant in cytostatic treatment, due to its potential synergistic effects in combination with these drugs.  

 Minor editing of English language required.

     Thank you for pointing it out. The English language has been revised.

Round 2

Reviewer 2 Report

Comments and Suggestions for Authors

My coments were addressed.

Reviewer 4 Report

Comments and Suggestions for Authors

The authors have addressed my comments and revised the original manuscript accordingly. Therefore I would like to recommend the current version for publication.